# Antibacterial Properties of TMA against *Escherichia coli* and Effect of Temperature and Storage Duration on TMA Content, Lysozyme Activity and Content in Eggs

**DOI:** 10.3390/foods11040527

**Published:** 2022-02-11

**Authors:** Xuefeng Shi, Xingzheng Li, Xianyu Li, Zhaoxiang He, Xia Chen, Jianlou Song, Lingsen Zeng, Qianni Liang, Junying Li, Guiyun Xu, Jiangxia Zheng

**Affiliations:** 1College of Animal Science and Technology, China Agricultural University, Beijing 100193, China; xuefeng.shi@cau.edu.cn (X.S.); xianyuli96@163.com (X.L.); ahknow@163.com (Z.H.); songjianlou@163.com (J.S.); s20203040561@cau.edu.cn (L.Z.); lqn17603727160@163.com (Q.L.); lijunying@cau.edu.cn (J.L.); ncppt@cau.edu.cn (G.X.); 2Shenzhen Agricultural Genome Research Institute, Chinese Academy of Agriculture Sciences, Shenzhen 440307, China; lxz2019@alu.cau.edu.cn; 3Institute of Animal Husbandry and Veterinary Medicine, Beijing Academy of Agriculture and Forestry Sciences, Beijing 100094, China; chenxia_91@163.com

**Keywords:** TMA, lysozyme, cuticle, bacteriostasis

## Abstract

Studies on trimethylamine (TMA) in egg yolk have focused on how it impacts the flavor of eggs, but there has been little focus on its other functions. We designed an in vitro antibacterial test of TMA according to TMA concentrations that covered the TMA contents typically found in egg yolk. The change in TMA content in yolk was analyzed at different storage temperatures and for different storage durations. The known antibacterial components of eggs, including the cuticle quality of the eggshell and the lysozyme activity and content in egg white, were also assessed. The total bacterial count (TBC) of different parts of eggs were detected. The results showed that the inhibitory effect of TMA on *Escherichia coli* (*E. coli*) growth increased with increasing TMA concentration, and the yolk TMA content significantly increased with storage duration (*p* < 0.05). The cuticle quality and lysozyme content and activity significantly decreased with storage time and increasing temperature, accompanied by a significant increase in the TBC on the eggshell surface and in the egg white (*p* < 0.05). This work reveals a new role for trace TMA in yolks because it reduces the risk of bacterial colonization, especially when the antibacterial function of eggs is gradually weakened during storage.

## 1. Introduction

Eggs are an inexpensive and high-quality source of nutrients, especially proteins, iron, vitamins, and phosphorus. Their availability, low cost, ease of preparation, taste, and low caloric value enable eggs to meet the nutritional needs of humans. Because of their high nutrient content, care must be taken to prevent eggs from becoming a source of bacteria and to ensure that their quality meets consumer requirements.

Over the course of evolution, avian eggs formed a set of physical and chemical antibacterial mechanisms to resist the invasion of external microorganisms. The physical defense mechanism is based on the eggshell and the chemical defense mechanism is based on antibacterial substances in the egg white and yolk [1]. The physical defense mechanism is mainly composed of the egg cuticle and eggshell. External microorganisms must first pass through the eggshell. The shell structure includes the inner and outer shell membranes, mammillae, palisade layers, vertical crystal layers, and cuticles [2]. The cuticle is a protective defense covering the outermost layer of the eggshell and plays an important role in resisting microorganism invasion [3]. The physical properties and chemical composition of egg cuticles reduce microbial contamination [4]. As the first barrier of an egg, the egg cuticle is in direct contact with the external environment and plays an important role in preventing microbial invasion [5,6]. 

The chemical defense mechanism mainly comprises antibacterial substances in the egg white and yolk. When the outer protective system of an egg becomes compromised, the internal chemical defense mechanism plays a role [7]. Among the chemical defense mechanisms, the most important are various antibacterial proteins, especially lysozyme [8]. Egg white contains different proportions of antibacterial substances, mainly ovotransferrin, lysozyme, ovomucoid, and ovoid inhibitors [9]. Ovotransferrin is effective against the *Bacillus cereus* group, and the effect is better under alkaline conditions [10]. Egg white contains high levels of lysozyme, accounting for approximately 3.5% of the entire egg white [11]. Lysozyme is a muramidase that can lyse the cell walls of microorganisms to achieve an antibacterial effect and has a strong inhibitory effect on Gram-positive bacteria [12]. In addition to the antibacterial substances in egg white, yolk immunoglobulin in egg yolk can destroy pathogens and inhibit the growth of bacteria [13]. In addition, yolk lipoprotein and yolk phosphatidic acid protein inhibit bacterial activity through lipid mediation [14].

The content of biogenic amines has been used as an indicator of quality for many foods, such as fish, meat, and eggs [15,16]. The quantity and type of amines in a food depend on the nature of the food and its microbiota [17]. To our knowledge, few studies have evaluated the biogenic amine content of eggs during their shelf life. Understanding these concentrations is important to improve the nutritional composition of eggs and evaluate their quality. TMA is found in egg yolk and is a type of biogenic amine that is mainly produced from choline in the bird’s feed through microbial action [18,19]. The content range of TMA in chicken egg yolks is 2 ~ 3 μg/g [19]. People perceive a fishy smell when the TMA concentration is greater than 4 μg/g [20,21].

Ward et al. showed that common eggs contain only trace amounts of TMA (less than 4 μg/g in yolk), and most people do not perceive a fishy smell from them [22]. Many studies on TMA in eggs have focused on its effect on egg flavor, but its other functions have received little attention. However, studies have shown that TMA can reduce the accumulation of plaque and has antibacterial and anti-inflammatory properties [23,24]. To date, no studies have investigated the antibacterial activity of TMA in eggs or its possible contribution to the chemical aspect of antimicrobial defense. 

The objective of the work was to investigate the antibacterial properties of TMA against *E. coli*. We also assessed changes in the TMA content, cuticle quality, and lysozyme content and activity in eggs under different storage temperature and duration conditions to further clarify changes in the antibacterial mechanisms of eggs.

## 2. Materials and Methods

### 2.1. Ethics Statement

All of the experimental procedures were conducted in accordance with the *Guidelines for Experimental Animals* established by the Animal Care and Use Committee of China Agricultural University (permit number: AW52501202-1-1). The animal studies were in compliance with the ARRIVE guidelines [25].

### 2.2. Determination of Antibacterial Activity of TMA against E. coli

The antibacterial activity of TMA was determined according to a protocol adapted from the two-Petri-dish assay [26]. TMA solutions with concentrations of 1, 2, 3, 4, and 5 μg/mL were prepared from TMA hydrochloride (Shanghai Macklin Biochemical Co., Ltd, Shanghai, China). The plates were incubated overnight at 37 °C, and colonies were counted and recorded. *E. coli* containing plasmid pGLO (BIO-RAD, Laboratories) were grown overnight with shaking at 37 °C in agarose nutrient liquid medium (10 g tryptone, 5 g yeast extract, and 10 g sodium chloride, Sangon, Shanghai, China) containing 100 μg/mL carbenicillin. Cultures were inoculated at a dilution of 1:50 in fresh L-broth containing 100 μg/mL carbenicillin and either 5 mM or 1 mM agarose nutrient liquid medium.

### 2.3. Egg Collection and Storage

The study used a laying-hen commercial diet with the code of 324 produced by Charoen Pokphand Group, Beijing, China. The hens were not given antibiotics in their feed, water, or by injection. A total of 600 eggs were produced by 36-week-old hens of the same strain (Dwarf) at the National Center of Performance Testing of Poultry (Beijing, China). The hens were housed in the same laying facility, and the procedures and diet used during the production period were the same for all birds. The eggs were collected at the same time. Shortly thereafter, the unwashed eggs were selected and classified and then randomly stored at 25 °C [10% RH] or 4 °C [65% RH] for 42 days. All eggs were placed on plastic trays to ensure that each egg was securely stored with adequate room. The initial sampling period was conducted 1 day after collection and each subsequent week for the designated time frame. Ten egg samples were randomly selected from each condition for analysis throughout the 7 weeks.

### 2.4. Determination of TMA Content in Yolk

The yolk TMA content was determined using ultraviolet spectrophotometry [27]. The absorbance change at 410 nm was measured by a UV–vis spectrophotometer (v-1800, Shimadzu, Tokyo, Japan). A TMA standard curve was used to determine the yolk TMA content, with the TMA standard solution concentration as the abscissa and the absorbance value as the ordinate.

### 2.5. Determination of Lysozyme Activity and Content

The lysozyme content was determined according to turbidimetry [28]. First, a lysozyme standard (Nanjing Jiancheng Bioengineering Institute, Nanjing, China) was dissolved in 0.90% NaCl solution, and various lysozyme standard concentrations were prepared. A standard curve was constructed based on the absorbance at λ = 450 nm using the UV–vis spectrophotometer. The egg white samples were diluted 400-fold, and the absorbance levels were recorded in the same way. The final results were calculated using the standard curve.

The lysozyme activity in egg white was assessed according to the spectrophotometric method of Kijowski and Leśnierowski [29]. A cell suspension was produced by adding lyophilized *Micrococcus lysodeikticus* cells (BeNa Culture Collection, BNCC138438, Beijing, China) to potassium phosphate buffer (pH 6.24) at 25 °C. The absorbance of the resulting suspension at λ = 450 nm was between 0.6 and 0.7. One milliliter of egg white sample from the treatment group was added to 3.0 mL of the M. lysodeikticus suspension and mixed by inversion. Changes in absorbance at 450 nm were monitored by UV–vis at 0, 1, and 3 min at 25 °C. The slope (A450 nm/min) of the linear portion of the curve for the absorbance at 450 nm vs. time was used to calculate the biological activity of the lysozyme in enzyme units (EU). A decrease of 0.001 × A450 nm/min was defined as 1 EU.

### 2.6. Determination of Cuticle Quality

The cuticle quality was determined according to the opacity method (α value, %) [6]. The cuticle quality was determined by MST cuticle blue (MS Technologies Ltd, London, UK) and a spectrophotometer (CM-2600d; Konica Minolta, Japan).

### 2.7. Determination of Total Bacterial Count (TBC)

Each egg was placed into one Whirl-Pak (Nasco, Silver Valley, ID, USA) bag containing 100 mL of sterile physiological saline (0.9% NaCl). A whole-egg washing method with a vortex was used to recover eggshell surface bacteria. The eggshell surface was thoroughly disinfected with ethanol (75%) and exposed to ultraviolet light for 30 min after treating the eggshell. The egg white and egg yolk were removed with a sterile syringe after opening the eggshell. The eggshell eluent was diluted at ratios of 1:100, 1:1000, and 1:10,000. Egg white or yolk was diluted at ratios of 1:1, 1:10, and 1:100 under sterile physiological conditions. A 200 μL sample dilution was plated on solid nutrient agar medium (10 g tryptone, 5 g yeast extract, 10 g sodium chloride, and 15 g agar/L). The plates were incubated overnight at 37 °C, and colonies were counted and recorded as colony-forming units (CFUs).

### 2.8. Statistical Analysis

The basic descriptive statistical results are shown as the mean and standard deviation (mean ± SD). All statistical analyses were performed using SPSS 25. The significance test was carried out by repeated measurement variance analysis. Statistically significant differences among treatments were determined by the least significant difference (LSD) test or *t*-test. The experimental results were plotted using GraphPad Prism 7.0 software.

## 3. Results and Discussion

### 3.1. Effect of TMA Concentration on E. coli Growth

The antibacterial experiment showed that TMA had antibacterial ability (Figure 1). There was no significant difference in the number of *E. coli* between the 1 μg/mL TMA group and the control group (*p* < 0.05). However, there was a significant difference in the number of *E. coli* between the 2, 3, 4, and 5 μg/mL TMA groups and the control group (*p* < 0.05). The antibacterial effect was observable because the number of *E. coli* colonies decreased as the TMA concentration increased (Figure 2).

The high TMA content in egg yolk is the main factor leading to the fishy taste of duck eggs and chicken eggs. The antibacterial effect of trace TMA in vitro was proven in the present study. This result was consistent with previous studies showing that TMA can reduce the survival rate of certain bacteria. Some volatile molecules can not only regulate bacterial antibiotic resistance but also have antifungal or antibacterial activity [30,31]. TMA is a particularly potent example of this effect. Jones et al. found that exposure to TMA increases the sensitivity of bacteria to aminoglycoside antibiotics and inhibits bacterial growth [24]. *Streptomyces*-produced TMA in soil has antibacterial activity. Jones found that, in soil, *Streptomyces*-produced TMA sensitizes bacteria to the effect of *Streptomyces*-produced antibiotics in addition to having direct antibacterial activity. TMA was shown to reduce the survival of soil bacteria (*Bacillus subtilis* and *Micrococcus luteum*) by increasing the pH of the growth medium from 7.0 to 9.5 [24]. An increase in pH will increase nutrient absorption and change the enzyme activity in the body, thus affecting bacterial growth [32].

The antibacterial and antifungal properties of TMA also appear to be related to its effect on regulating nutrition. The effective utilization of environmental iron, an essential nutrient for microbial growth, affects the survival of bacteria and fungi. Bacteria facilitate iron acquisition by releasing iron-chelating siderophores [33]. These iron-chelating siderophores bind ferric iron, and the resulting siderophore-iron complexes are brought into the cell through dedicated membrane transporters, after which the iron is released in the form of ferrous iron. The release of TMA increases the environmental pH. In alkaline environments, ferric iron and hydroxide ions form stable complexes, which reduces the combination of iron carriers and ferric ions and reduces the level of bioavailable iron, thus creating an uninhabitable environment for many bacteria [34,35,36]. Jones et al. confirmed this in 2019. The authors found that TMA released by *Streptomyces* explorer cells significantly changed the pH of their surrounding environment and reduced the survival of nearby soil microorganisms by decreasing the availability of iron [37]. Meswak (a “chewing stick”) refers to the roots or twigs of *Salvadora persica* L. and shows pharmacological activity and strong antibacterial activity [38,39]. A pharmacological study on meswak found that its chemical composition contained a large amount of TMA and that the TMA reduced plaque accumulation and had antibacterial activity and anti-inflammatory activity [40].

The antibacterial properties of TMA may have an important protective effect on breeding eggs. The antibacterial properties of TMA also reflect the evolutionary adaptability of ducks to survival and reproduction. Previous studies have shown that the TMA content of duck and goose eggs is higher than that of chicken eggs [41,42]. Compared to the breeding environment of chickens, ducks and geese usually lay their eggs in wetter ponds or grasslands. However, humid environments are prone to breeding various pathogenic microorganisms that threaten the incubation process [43]. Thus, the high TMA content might protect embryo development in duck and geese eggs.

In addition, the antibacterial activity of TMA in egg yolk is of great significance to the laying hen industry in terms of safety and pathogen control. Usually, some pathogens will enter the ovary with the hen’s reproductive system and become part of the egg, which will eventually be laid as the egg is formed. Our results showed that TMA is antibacterial against *E. coli*. Although we have not yet confirmed that TMA has antibacterial properties against other pathogens, it also provides suggestions for us to ensure the safety of egg quality. For example, we can reduce pathogen contamination and reduce morbidity in offspring by increasing the content of TMA in eggs.

### 3.2. Effects of Temperature and Storage Duration on TMA Content and TBC in Egg Yolks

Changes in the TMA content in egg yolks are shown in Table 1 and Figure 3A. There was a significant increase in the egg yolk TMA content during storage (*p* < 0.05). The TMA content increased from 2.369 μg/g to 2.989 μg/g at 25 °C and from 2.369 μg/g to 2.995 μg/g at 4 °C. However, the TMA contents in the two storage temperatures were not significantly different (*p* < 0.05). Moreover, no microorganisms were found in the yolks throughout the experiment.

TMA is a kind of biogenic amine. There are many common biogenic amines in eggs, including TMA, spermidine, putrescine, ethanolamine, and ethylamine [44,45]. The quantity and type of amines in food depend on the nature of the food and its microbiota [17]. Usually, TMA in egg yolk mainly comes from choline and other substances in the diet through the action of microorganisms, resulting in TMA deposition in the egg yolk [45]. However, we found no microorganisms in the egg yolk during the whole experiment. We also learned that biogenic amines are small molecules that are formed through decarboxylation of amino acids and through biosynthesis and enzymatic processes [16]. The increase of TMA in egg yolk might form through the processes above.

Figueredo et al. determined the content of biogenic amines in egg yolks at room temperature and refrigeration and found that the storage temperature did not affect the levels of biogenic amines [46]. In our study, there was also no significant difference in the TMA levels between the two storage temperatures (*p* < 0.05). The results are consistent with a previous study showing that storage temperature had no significant effect on the content of amines (*p* > 0.05) [17]. We confirmed that the TMA concentration in egg yolks was sufficiently high to have antibacterial activity and that the TMA content increased with the extension of storage time. Previous studies have shown that the TMA content in egg yolks is significantly higher than that in egg whites [42]. Microorganisms were detected in the egg white, but no microorganisms were found in the yolk. With the extension of storage time, the antibacterial effect of other antibacterial substances in eggs, such as lysozyme, was weakened; however, the effect of TMA was increased, indicating a change in the antibacterial substances in eggs. Therefore, the results showed that increasing the TMA content below the flavor threshold might help to improve the bacteriostasis of egg yolks.

### 3.3. Effects of Temperature and Storage Duration on the Content and Activity of Lysozyme and TBC in Egg White

The content and activity of lysozyme are an important part of the antibacterial system during egg storage, and lysozyme accounts for 3–4% of the total egg white protein [47]. With the extension of storage time, the content and activity of lysozyme showed downward trends (Table 1, Figure 3B,C). At 25 °C, the lysozyme content decreased from 6.296 mg/g to 4.800 mg/g; at 4 °C, the lysozyme content decreased less, from 6.296 mg/g to 5.589 mg/g. Thus, the lysozyme contents in cold storage and at room temperature did not significantly change, but the decrease at room temperature was greater than that in cold storage.

The lysozyme activity test showed that the lysozyme activity decreased at both storage temperatures. By the 42nd day of storage, the lysozyme activity was reduced by half compared to the first day in both cold storage and at room temperature. The activities at 25 °C and 4 °C both decreased from 26,592 U/mg at the beginning to 17,561 U/mg (*p* < 0.001) and 20,648 U/mg (*p* < 0.001) at the end of storage, respectively. As the lysosome content and activity decreased, the number of bacteria in the egg whites increased (Figure 4). There was no bacterial growth in the egg whites before 21 days in either storage condition. Bacterial growth was detected in egg white for the first time at 25 °C on the 21st day. A small number of bacteria were detected at 4 °C starting on the 35th day.

There are several methods to determine the content of lysozyme, mainly including turbidimetric and HPLC. Turbidimetric method has the advantages of simple operation, high sensitivity, but poor repeatability. HPLC method has the advantages of high sensitivity, good stability and good repeatability. In this experiment, the simple turbidimetric method was used to determine the content of lysozyme in egg white [48]. The lysozyme content and activity results were consistent with those of a previous study showing that they gradually decreased over time [49]. Board reported that there was a 20-day lag between the time an organism penetrated the eggshell and the detection of a large number of organisms in the eggshell [50]. The significant increase in the number of bacteria in egg white coincides with the significant decrease in lysozyme content and activity. The decreased lysozyme content and activity led to a decrease in the egg’s antibacterial defense system and an increase in the rate of bacterial invasion [51].

Lysozyme is a hydrolytic enzyme with the ability to degrade bacterial cell walls. It constitutes a mechanism of natural, primary protection for the embryo developing within the egg [52]. Lysozyme has lytic activity toward peptidoglycan that forms the cell wall of Gram-positive bacteria [53]. The extension of storage time changes the spatial structure or environment of lysozyme, which affects its activity [54]. The activity of lysozyme is also affected by the pH value in egg white. The optimum pH of lysozyme is 5.3 ~ 6.4; this pH range is used for high-acid food preservation [55]. Previous studies have shown that egg white viscosity, lysozyme and pH all inhibit bacterial reproduction in eggs [56]. However, with the extension of storage, the content and activity of lysozyme decreases, which increases the risk of bacterial invasion of poultry eggs.

### 3.4. Effects of Temperature and Storage Duration on Egg Cuticle Quality and TBC on Eggshell Surface

As the outermost layer of the eggshell, the cuticle has good hydrophobicity, which effectively reduces the contact area of droplets on the eggshell surface and keeps the surface dry, thus reducing the probability of eggs being contaminated by pathogenic microorganisms [57]. With the extension of the storage time, the quality of the cuticle gradually decreased (Table 1 and Figure 3D). Under storage conditions at 4 °C and 25 °C, the quality of the cuticle decreased from 28.7% to 24.3% and 23.2%, respectively. During storage, independent of the storage temperature (*p* > 0.05), there was a significant decrease in the quality of the cuticle (*p* < 0.05). In the first 7 days of storage, the cuticle quality decreased synchronously at both temperatures. From the 7th to the 21st day, the cuticle quality at 25 °C decreased significantly compared with the 4 °C storage. From the 21st to the 42nd day, the cuticle quality showed a significant downward trend. The number of bacteria on the eggshell increased with the decline in cuticle quality (Figure 5). During the first three weeks of storage, there was no significant difference in the bacteria number on the eggshell surface between the eggs stored at 4 °C and 25 °C. On the 7th and 14th days, the number of bacteria at 25 °C was greater than that at 4 °C; on the 21st day, the number of bacteria at 25 °C and 4 °C began to show a significant difference (*p* < 0.001). During the first two weeks of storage, the TBC at 25 ℃ and 4 ℃ increased from an initial level of 3.815 ± 0.060logCFU/egg to 4.087 ± 0.033logCFU/egg and 4.018 ± 0.057logCFU/egg, respectively, and there were no significant differences between the two temperatures, although the TBC at 25 ℃ was higher than that at 4 ℃. At 21 days, there was a clear difference. By 42 days, the TBC at 25 °C and 4 °C had increased to 5.150 ± 0.021logCFU/egg and 5.001 ± 0.010logCFU/egg, respectively, which were significant increases compared with the initial TBC (3.815 ± 0.060 logCFU/egg). Thus, with the extension of storage time, the number of bacteria on the eggshell surface increased at both 25 °C and 4 °C.

The storage time can affect the TBC on the surface of eggshells—the longer the storage time, the greater degree of contamination [58]. Our results showed that compared with the 25 °C storage, the 4 °C storage was less conducive to bacterial reproduction. The natural antibacterial barrier of eggs tends to decrease with storage time [59]. The number of microorganisms on the eggshell surface and in the egg white after 21 days of storage further supported this finding. The TBC on the eggshell surface increased slowly before day 21 and increased rapidly thereafter. The reason for this change is related to the effects of environmental temperature and storage time on microorganism growth [60]. At the same time, the cuticle quality decreased with the increase in bacteria, which indicated that bacteria influenced the cuticle quality. Olivier et al. proved that bacteria and fungi on the surface of an eggshell will gradually dissolve the eggshell cuticle, which reduces the coverage and quality of the eggshell cuticle [61]. This explains why the cuticle quality decreased significantly with a significant increase in the number of bacteria on the eggshell surface after the 21st day.

The cuticle plays an important role in resisting bacterial invasion [6]. The cuticle is a natural barrier against bacterial cross-shell invasion, and it can reduce bacterial counts through physical and chemical mechanisms [62]. With the extension of the storage time, the cuticle quality decreases, thereby increasing the risk of bacterial invasion [63]. Similar results were obtained in the current study, where the cuticle quality was significantly reduced under the two storage temperatures with the extension of the storage period. Although there were differences in humidity between the two storage conditions, it is more likely that the cuticle quality was impacted by the temperature rather than the humidity; this was shown in a previous study [64]. When the environmental temperature is high, the egg cuticle quality declines because it can more easily dry and fall off [65].

### 3.5. Other Chemical Defense Mechanisms in the Egg White and Yolk

In addition to lysozyme, the chemical defense mechanisms in egg whites are various antimicrobial proteins and protease inhibitors. Ovotransferrin in antimicrobial proteins can inhibit some Gram-negative bacteria, and lysozyme can inhibit most Gram-positive bacteria [66]. The functions of antimicrobial proteins are divided into two categories: degrading microbial components and chelating vitamins or minerals necessary for microbial growth. Protease inhibitors mainly include cysteine protease inhibitor, ovomucin, and egg inhibitor [67]. These three protease inhibitors can achieve the purpose of bacteriostasis by inhibiting the activity of various proteases. In addition, some glycoproteins and antimicrobial peptides in egg white also play a defensive role against the invasion of microorganisms [8]. 

Many components in egg yolk also showed antibacterial activity, mainly including yolk immunoglobulin, phosvitin, and lipovitellin [14]. Zhen et al. [68] showed that yolk immunoglobulin can effectively treat dairy cow mastitis in vitro, and can inhibit the growth of *E. coli* in vitro. Choi et al. [69] showed that the reason why lecithin high phosvitin has bactericidal property is that the structural stability of the bacterial cell membrane is reduced under heat treatment, and lecithin high phosvitin can chelate metal ions outside the cell membrane, resulting in cell membrane damage and the leakage of genetic material in the cell, resulting in bacterial death. Brady et al. [14] found that crude lipase or the combination of lipase and protease could release antibacterial activity after extracting lipovitellin in vitro. Therefore, vitellin is an important molecule of lipid mediated antibacterial activity, especially against *Pathogenic Streptococcus*.

## 4. Conclusions

The antibacterial activity of TMA in egg yolk was detected here for the first time. The deposition of TMA in egg yolks forms part of the antibacterial system in eggs together with other well-known substances such as lysozyme. The TMA content also reflects the replacement of antibacterial substances in eggs as the storage time is extended. This study deepens our understanding of the antibacterial components of eggs. Moreover, we can increase or decrease the content of TMA in egg yolk according to different production purposes. For example, we reduce the risk of bacterial invasion by increasing the content of TMA in breeding eggs, thereby reducing the probability of some vertically transmitted diseases.

## Figures and Tables

**Figure 1 foods-11-00527-f001:**
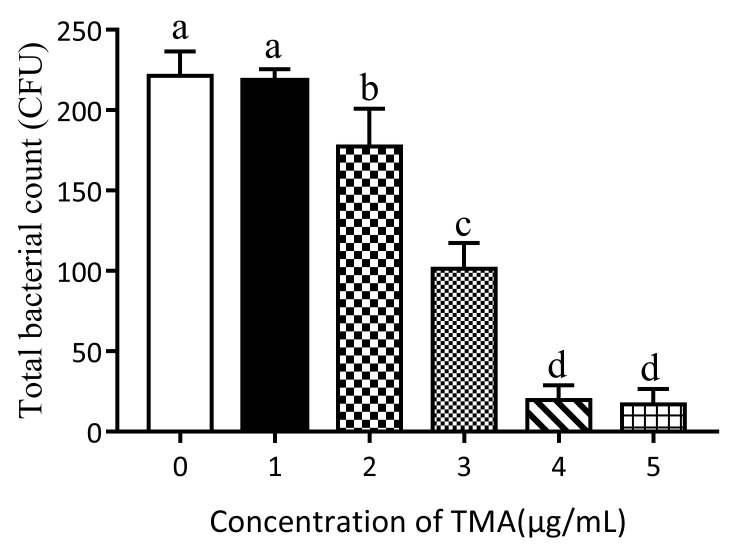
Total colony counts with TMA concentrations from 0 μg/mL to 5 μg/mL after 24 h at 37 °C. Different letters (a, b, c, d) indicate a significant difference (*p* < 0.05).

**Figure 2 foods-11-00527-f002:**
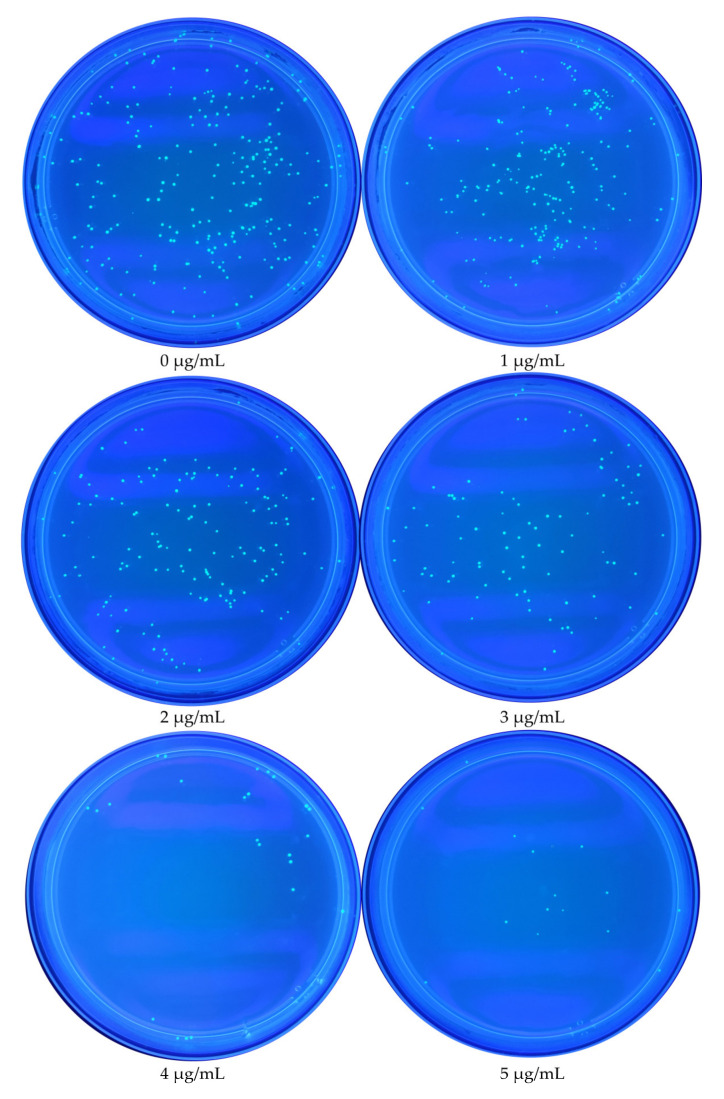
Schematic of the plate-based assay used to assess the effect of TMA on *E. coli* colonies.

**Figure 3 foods-11-00527-f003:**
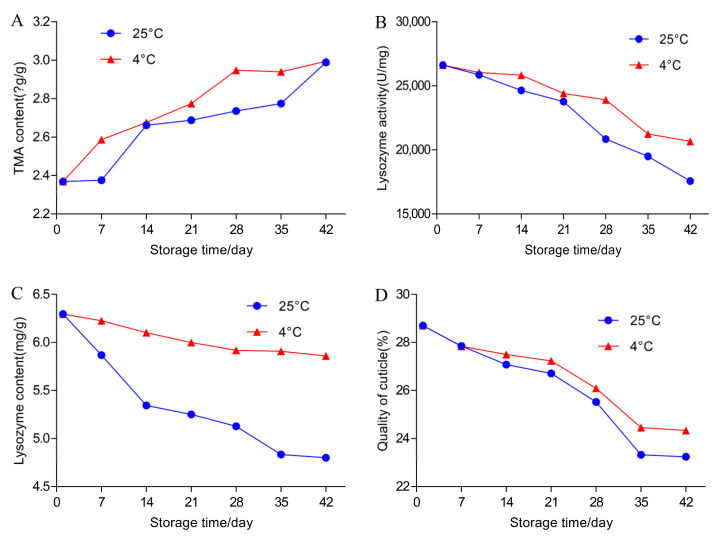
Effect of different storage temperatures and extended storage on the TMA content (**A**), lysozyme activity (**B**), lysozyme content (**C**), and cuticle quality (**D**).

**Figure 4 foods-11-00527-f004:**
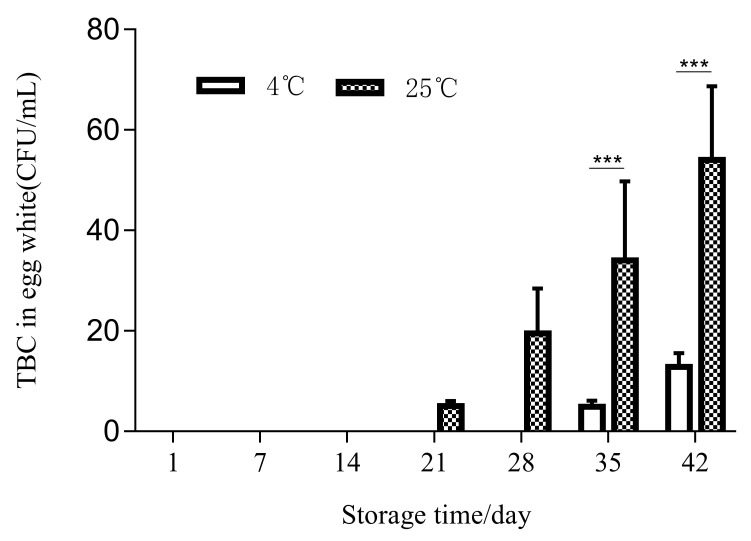
The TBC in egg white changed depending on the storage temperature and storage time (*** *p* < 0.001).

**Figure 5 foods-11-00527-f005:**
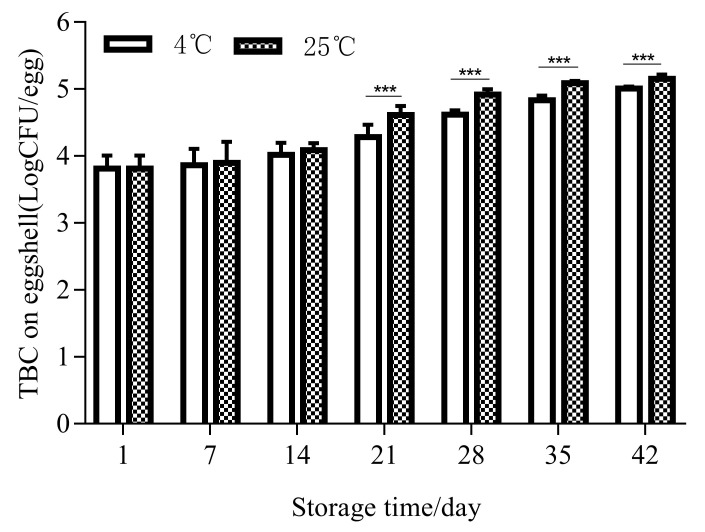
The TBC on the egg eggshell changed with different storage temperatures and storage times (*** *p* < 0.001).

**Table 1 foods-11-00527-t001:** Effect of storage temperature and time on different egg indices for up to 42 days of storages.

Index	T (°C)	Storage Time (Day)						
		1	7	14	21	28	35	42
TMA content (μg/g)	25 4	2.369 ± 0.186 ^c^ 2.369 ± 0.186 ^c^	2.375 ± 0.68 ^c,y^ 2.587 ± 0.161 ^c,x^	2.662 ± 0.15 ^b^ 2.677 ± 0.146 ^bc^	2.688 ± 0.394 ^b^ 2.774 ± 0.127 ^b^	2.736 ± 0.568 ^b,y^ 2.948 ± 0.121 ^a,x^	2.775 ± 0.166 ^b,y^ 2.989 ± 0.043 ^a,x^	2.989 ± 1.116 ^a^ 2.995 ± 0.044 ^a^
lysozyme activity (×10^5^ U/mg)	25 4	2.662 ± 0.119 ^a^ 2.662 ± 0.119 ^a^	2.586 ± 0.077 ^b,y^ 2.605 ± 0.106 ^a,x^	2.464 ± 0.083 ^c,y^ 2.584 ± 0.063 ^b,x^	2.377 ± 0.138 ^d,y^ 2.440 ± 0.043 ^b,x^	2.084 ± 0.145 ^d,y^ 2.391 ± 0.135 ^c,x^	1.949 ± 0.839 ^e,y^ 2.124 ± 0.112 ^c,x^	1.756 ± 0.029 ^e,y^ 2.066 ± 0.086 ^c,x^
lysozyme content (mg/g)	25 4	6.296 ± 0.215 ^a^ 6.296 ± 0.215 ^a^	5.867 ± 0.065 ^a,y^ 6.226 ± 0.183 ^ab,x^	5.344 ± 0.134 ^b,y^ 6.103 ± 0.141 ^bc,x^	5.251 ± 0.139 ^c,y^ 6.000 ± 0.193 ^cd,x^	5.128 ± 0.097 ^c,y^ 5.918 ± 0.155 ^d,x^	4.833 ± 0.045 ^d,y^ 5.908 ± 0.198 ^d,x^	4.800 ± 0.088 ^e,y^ 5.589 ± 0.098 ^d,x^
quality of cuticle (%)	25 4	28.701 ± 1.894 ^a^ 28.701 ± 1.894 ^a^	27.840 ± 2.337 ^a^ 27.853 ± 0.926 ^ab^	27.077 ± 2.206 ^ab,y^ 27.494 ± 2.610 ^ab,x^	26.708 ± 1.289 ^ab,y^ 27.231 ± 1.82 ^ab,x^	25.524 ± 1.612 ^b,y^ 26.092 ± 2.142 ^bc,x^	23.325 ± 0.894 ^c,y^ 24.460 ± 1.676 ^c,x^	23.244 ± 1.406 ^c,y^ 24.337 ± 1.299 ^c,x^

^a,b,c,d,e,f^ means within a row that do not share a common superscript differ significantly (*p* < 0.05). ^x,y^ means within a column of the same that do not share a common superscript differ significantly (*p* < 0.05).

## Data Availability

The data that support the findings of this study are available on the request from the corresponding author. The data are not publicly available due to privacy or ethical restrictions.

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
