# Peer review of "Antibacterial Properties of TMA against Escherichia coli and Effect of Temperature and Storage Duration on TMA Content, Lysozyme Activity and Content in Eggs"

_foods, 2022, doi:10.3390/foods11040527_

Round 1

Reviewer 1 Report

The subject of this article is interesting. However, there are few observations which may help improving the manuscript.

  1. I will also suggest a broad series of egg pathogens instead of using only coli and Micrococcus lysodeikticus. However, the use of already used pathogens has not been advocated in introduction and methods sections. I
  2. I will suggest using Salmonella enterica like Typhimurium and Enteritidis to be included in the study design along others, because they happen to be one of the key pathogens from egg products.
  3. Why 5ug/mL concentration has been used, when studies suggest than a concentration of less than 4.5 or sometimes 4 is acceptable in terms of taste and flavor.
  4. Authors can elaborate the practical implications of these findings for industry in terms of safety and pathogen control.
  5. The introduction lacks necessary details about the subject matter on study design including choice of pathogen used in study. The purpose to perform some experiments remain unclear like why lysozyme content and activity was monitored.
  6. The data illustration can be improved by using a Figure instead of a long Table 1.
  7. I will suggest that results section be separate from the discussion section. The results have not been discussed in sufficient detail.
  8. The methods lack the clarity and flow. Most methods sections leave necessary details like strain Micrococcus lysodeikticus strain ID can be provided. The reason to choose this pathogen has not been discussed in introduction and discussion. It remains unclear why this pathogen was used along with coli. The information on the type and source of E. coli used is not provided. Why other pathogens such as Salmonella Typhimurium, Salmonella Enteriditis were not selected instead of E. coli and Micrococcus lysodeikticus.
  9. The study design has completely ignored the potential antimicrobial effect of antibiotic residues in the eggs. Since, no details about the egg production process are given. We expect generally some levels of antibiotic residues in commercially produced eggs, which can inhibit microbes in the product. The reasons may be elaborated.
  10. There is no mention of E. coli in introduction and methods, while we see mention of E. coli in results.
  11. The language in manuscript may be improved in general.

Author Response

Dear reviewers:

Thank you for your comments concerning our manuscript entitled “Antibacterial properties of TMA and effect of temperature and storage duration on TMA content in egg yolk” (ID: foods-1572810). Those comments are all valuable and very helpful for revising and improving our paper, as well as the important guiding significance to our researcher. We have studied comments carefully and have made a correction which we hope meet with approval. Revised portions are marked up using the “Track Changes” on the manuscript. The main correction in this manuscript and the responses to reviewer’s comments are below:

Response to Reviewer 1 Comments:

-------------------------------------------------------------------

The subject of this article is interesting. However, there are few observations which may help improving the manuscript.

Major comments:

Comment #1:

  1. I will also suggest a broad series of egg pathogens instead of using only coli and Micrococcus lysodeikticus. However, the use of already used pathogens has not been advocated in introduction and methods sections.

Response 1: Done. In the materials and methods, we added the strains used to evaluate antibacterial activity. Please see line 94-99.

Micrococcus lysodeikticus is a strain that we use to determine the content of lysozyme in egg white, not a pathogen. Please see “2.5. Determination of Lysozyme Activity and Content”.

In addition, in order to make the combination of article title and content more perfect, we have revised the article title to “Antibacterial properties of TMA against Escherichia coli and effect of temperature and storage duration on TMA content, lysozyme activity and content in eggs”. Please see line 1-3.

Comment #2:

  1. I will suggest using Salmonella enterica like Typhimurium and Enteritidis to be included in the study design along others, because they happen to be one of the key pathogens from egg products.

Response 2: Thank you for your advice.

We selected E. coli as the pathogen by referring to the following literature:

[1] Ko K Y ,  Mendoncam A F ,  Ismail H , et al. Ethylenediaminetetraacetate and lysozyme improves antimicrobial activities of ovotransferrin against E. coli O157:H7[J]. Poultry Science, 2009, 88(2):406.

[2] Khan M S ,  Nakamura S ,  Ogawa M , et al. Bactericidal action of egg yolk phosvitin against Escherichia coli under thermal stress.[J]. Journal of Agricultural & Food Chemistry, 2000, 48(5):1503-6.

Certainly, your advice is also the direction we are about to study: whether TMA has antibacterial activity against other pathogens and the practical application of TMA's antibacterial properties in production.

Thanks again. Please see line 210-217.

Comment #3:

  1. Why 5μg/mL concentration has been used, when studies suggest than a concentration of less than 4.5 or sometimes 4 is acceptable in terms of taste and flavor.

Response 3: Thank you for your comments.

We set the concentration gradient of trimethylamine according to the content range of TMA in egg yolk(2~3μg/g), and also comprehensively consider the problem of fishy smell.

Comment #4:

  1. Authors can elaborate the practical implications of these findings for industry in terms of safety and pathogen control.

Response 4: Thank you for your advice.

We have added relevant discussion sections. Please see line 210-217.

Comment #5:

  1. The introduction lacks necessary details about the subject matter on study design including choice of pathogen used in study. The purpose to perform some experiments remain unclear like why lysozyme content and activity was monitored.

Response 5: Done. Please see line 80-83.

The design idea of our article:

Point 1: Antibacterial effect of trace trimethylamine (TMA) in egg yolk

The present study is based on the article published by our team in JFS in 2018 [Li, X., Yuan, G., Chen, X., Guo, Y., Yang, N., Pi, J., Zhang, H., & Zheng, J. (2018). Fishy Odor and TMA Content Levels in Duck Egg Yolks. Journal of food science, 83(1), 39-45.]. We found that the fishy smell of duck eggs was caused by TMA, and there was also trace TMA in other eggs. This makes us think, since TMA has an unpleasant smell, what is the necessity of TMA existence in egg yolk? We speculate that trace TMA may have antibacterial effect. Therefore, we designed the experiments and verified that trace TMA in egg yolk has an antibacterial effect without affecting the egg flavor. 

Point 2: The content of TMA in egg yolk increased with the extension of storage period.

From cuticle to lysozyme, studies had shown that eggs have a complete self-antibacterial system. In the process of studying TMA in eggs, we found that with the extension of storage time, the defense system of cuticle and lysozyme gradually weakened, while the content of TMA gradually increased (from 2.369μg/g to 2.989μg/g at 25 °C and from 2.369μg/g to 2.995μg/g at 4 °C.). Combined with the research conclusion of above point 1, we believe that TMA in egg yolk and other well-known substances (such as lysozyme) constitute whole antibacterial system from yolk to shell. When the storage time is prolonged and other antibacterial barriers of eggs are gradually weakened, the content of TMA increases and it continues to play an antibacterial role, which is of great significance to the safety of eggs. This has not been reported in the previous literature.

At the end of the introduction, we added the goal of this work and explained why to monitor the content and activity of lysozyme.

Comment #6:

  1. The data illustration can be improved by using a Figure instead of a long Table 1.

Response 6: Thank you for your comments.

In order to better display our test results, we added corresponding data Figure. Please see line224-227

Comment #7:

  1. I will suggest that results section be separate from the discussion section. The results have not been discussed in sufficient detail.

Response 7: Thank you for your advice.

We have tried to separate the two parts according to your advice. We found that after separating the results from the discussion, the combination of logic and content is not as perfect as before. So we finally put the two parts together. Thank you for your understanding. In order to ensure that the results and discussion are more complete, we have added some sufficiently detailed discussions in the results and discussion section. We also added section 3.5. Please see lines 210-217,233-241, 278-289, and 353-374.

Comment #8:

  1. The methods lack the clarity and flow. Most methods sections leave necessary details like strain Micrococcus lysodeikticusstrain ID can be provided. The reason to choose this pathogen has not been discussed in introduction and discussion. It remains unclear why this pathogen was used along with coli. The information on the type and source of  coli used is not provided. Why other pathogens such as Salmonella Typhimurium, Salmonella Enteriditis were not selected instead of E. coli and Micrococcus lysodeikticus.

Response 8: Thank you for your comments. Done。 Please see line 80-83, 94-99 and 210-217.

We have added Micrococcus lysodeikticus strain ID. Please see line 125. Micrococcus lysodeikticus is a strain that we use to determine the content of lysozyme in egg white, not a pathogen. Please see “2.5. Determination of Lysozyme Activity and Content”. We have added information on the type and source of E. coli used.

Comment #9:

  1. The study design has completely ignored the potential antimicrobial effect of antibiotic residues in the eggs. Since, no details about the egg production process are given. We expect generally some levels of antibiotic residues in commercially produced eggs, which can inhibit microbes in the product. The reasons may be elaborated.

Response 9: Thank you for your comments.

All test samples were taken from the National Center of Performance Testing of Poultry (Beijing, China). We participated in the whole rearing process of laying hens. The study used laying-hen commercial diet with the code of 324 produced by Charoen Pokphand Group, China. The hens were not given antibiotics in their feed, water or by injection. Please see line 101-103.

Comment #10:

  1. There is no mention of E. coli in introduction and methods, while we see mention of E. coli in results.

Response 10: Thank you for your comments.

Done. In the materials and methods, we added the strains used to evaluate antibacterial activity. Please see line 94-99.

Comment #11:

  1. The language in manuscript may be improved in general.

Response 11: Thanks. We have carefully edited the entire manuscript and the manuscript has been polished by a professional company. The proof of English polishing was attached.

Sincerely,

Dr. Jiangxia Zheng and Dr. Xuefeng Shi

Reviewer 2 Report

After reading the work titled:
 Antibacterial properties of TMA and effect of temperature and 2 storage duration on TMA content in egg yolk. I consider that it needs certain changes before being considered for publication by the Journal.

Throughout the manuscript there is an inappropriate use of capital letters, for example, line 15: Abstract: Studies on Trimethylamine (TMA) should be changed to Abstract: studies on trimethylamine (TMA). Please check this in the manuscript especially in the titles.
Line 16: We designed the in vitro antibacterial. The word in vitro must be in italics, please review the entire manuscript because this occurs every time this word is written.
Line 22: E. coli growth increased… the first time the microorganism is named, the scientific name must be complete. Escherichia coli
Lines 36-59: this paragraph is very long and should be separated or summarized so that it is not boring for the readers.
At the end of the introduction it should be clear what the main objective of the work was. For example, the objective of the work was…
Line 117: Micrococcus lysodeikticus… scientific names of microorganisms should be in italics. Please correct this throughout the manuscript.
In materials and methods it is not indicated which strain of bacteria was used to evaluate the antibacterial activity, please indicate it. Later you can see that only the antibacterial activity against E. coli was measured. It seems for me a somewhat poor study, different gram-negative and positive bacteria should be used. There is only one gram-negative organism. The title states antibacterial activity and implies that this TMA molecule has activity against many bacteria and this has not been proven. So the title should be changed for that reason and be more specific.
I have also observed another quite important problem in the antibacterial activity tests and the absence of positive control (antibiotic) that helps us confirm the normal behavior of the bacteria. Negative control was also not used.
The lysozyme content was calculated using a turbidimetric method. This method has its drawbacks as it should have been calculated by a more specific HPLC method.
Establish a single criterion to express numerical data, Line 235: 6.3 mg/g to 4.7 mg/g; a single decimal is used, line 287: to 4.087±0.033logCFU/egg and 4.018±0.057logCFU/egg three decimals are used.

Author Response

Dear reviewers:

Thank you for your comments concerning our manuscript entitled “Antibacterial properties of TMA and effect of temperature and storage duration on TMA content in egg yolk” (ID: foods-1572810). Those comments are all valuable and very helpful for revising and improving our paper, as well as the important guiding significance to our researcher. We have studied comments carefully and have made a correction which we hope meet with approval. Revised portions are marked up using the “Track Changes” on the manuscript. The main correction in this manuscript and the responses to reviewer’s comments are below:

Response to Reviewer 2 Comments:

-------------------------------------------------------------------

Dear Authors 

After reading the work titled:
Antibacterial properties of TMA and effect of temperature and 2 storage duration on TMA content in egg yolk. I consider that it needs certain changes before being considered for publication by the Journal.

Major comments:

Comment #1:
Throughout the manuscript there is an inappropriate use of capital letters, for example, line 15: Abstract: Studies on Trimethylamine (TMA) should be changed to Abstract: studies on trimethylamine (TMA). Please check this in the manuscript especially in the titles.
Response 1: Done. Please see line 16.

Comment #2:

Line 16: We designed the in vitro antibacterial. The word in vitro must be in italics, please review the entire manuscript because this occurs every time this word is written.

Response 2: Done. Please see line17, 171, 366, 366 and 373.

Comment #3:

Line 22: E. coli growth increased… the first time the microorganism is named, the scientific name must be complete. Escherichia coli

Response 3: Done. Please see line 23.

Comment #4:

Lines 36-59: this paragraph is very long and should be separated or summarized so that it is not boring for the readers.

Response 4: Done. Please see line 38-63.

Comment #5:
At the end of the introduction it should be clear what the main objective of the work was. For example, the objective of the work was…

Response 5: Done. Please see line 80-83.

Comment #6:
Line 117: Micrococcus lysodeikticus… scientific names of microorganisms should be in italics. Please correct this throughout the manuscript.

Response 6: Done. Please see line 56, 127, 177, 178, 180, and 194.

Comment #7:
In materials and methods it is not indicated which strain of bacteria was used to evaluate the antibacterial activity, please indicate it. Later you can see that only the antibacterial activity against E. coli was measured. It seems for me a somewhat poor study, different gram-negative and positive bacteria should be used. There is only one gram-negative organism. The title states antibacterial activity and implies that this TMA molecule has activity against many bacteria and this has not been proven. So the title should be changed for that reason and be more specific.

Response 7:

Done. In the materials and methods, we added the strains used to evaluate antibacterial activity. Please see line 94-99.

As your see, only one gram-negative bacteria, strains of E. coli contain the plasmid pGLO (Bio-Rad Laboratories), was selected for antibacterial test, because it is suitable for our laboratory with Biosafety level 1. According to your comments, the title has been changed to “Antibacterial properties of TMA against Escherichia coli and effect of temperature and storage duration on TMA content, lysozyme activity and content in eggs”.

Comment #8:
I have also observed another quite important problem in the antibacterial activity tests and the absence of positive control (antibiotic) that helps us confirm the normal behavior of the bacteria. Negative control was also not used.

Response 8: Thank you for your comments. The present experimental design is referred to the following paper:Bernier, S.P.; Letoffe, S.; Delepierre, M.; Ghigo, J.M. Biogenic ammonia modifies antibiotic resistance at a distance in physically separated bacteria. Mol. Microbiol. 2011, 81, 705-716.

In our study, the control group, trimethylamine concentration = 0 μg/mL, was set to help us confirm the normal behavior of bacteria. Please see line161 to 165.

Comment #9:
The lysozyme content was calculated using a turbidimetric method. This method has its drawbacks as it should have been calculated by a more specific HPLC method.

Response 9: Thank you very much for your advice. For the present study, the turbidimetric method was used to calculated the lysozyme content as the description in following paper: Lewko, L.; Krawczyk, J.; Calik, J. Effect of genotype and some shell quality traits on lysozyme content and activity in the albumen of eggs from hens under the biodiversity conservation program. Poult. Sci. 2021 100, 100863. https://doi.org/10.1016/j.psj.2020.11.040.

Limitation of the turbidimetric method has been discussed in line 278-289.

Now we know the HPLC method is more specific, and it will be used in our future studies. Thanks again.

Comment #10:
Establish a single criterion to express numerical data, Line 235: 6.3 mg/g to 4.7 mg/g; a single decimal is used, line 287: to 4.087±0.033 logCFU/egg and 4.018±0.057 logCFU/egg three decimals are used.

Response 10: Done. Please see line 258-265.

Sincerely,

Dr. Jiangxia Zheng and Dr. Xuefeng Shi

Reviewer 3 Report

Dear Authors,

the present manuscript ("Antibacterial properties of TMA and effect of temperature and storage duration on TMA content in egg yolk ") is an important and actual work. It is in a frame of the major journal scopes. I do not doubt the technical quality of the work and feel that there is a sufficient impact on a broader readership.

It will be very useful, if the authors can provide some corrections in the text and more extended discussions of the presented experiments. In particular,  

  1. It is important to correct a title in the following ways: 1) to use “Trimethylamine” instead of “TMA”; 2) there are many data on lysozyme activity and content (in addition to the TMA content in egg yolk) that must be addressed in the title too (i.e. “lysozyme activity”).
  2. It is important to divide Figure 1. (A) and (B) on the separate Figures 1 and 2 in order to increase the size ( at least in 3 times) of the photos “(B) Schematic of the 156 plate-based assay used to assess the effect of TMA on E. coli colonies”. It will be a better indication of these significant difference.
  3. The part “3.2. Effects of temperature and storage duration on TMA content and TBC in egg yolks” (lines 198-203). It is important to discuss the obtained TMA differences (“Changes in the TMA content in egg yolks are shown in Table 1”). It is especially important to discuss the “increased from 2.369 μg/g to 2.989 μg/g at 25°C and from 2.369 μg/g to 2.995 μg/g at 4°C” (lines 199-201), because the authors insisted that “no microorganisms were found in the yolks throughout the experiment” (lines 202-203). Usually, such changes in the TMA content in eggs must be due to the action of microorganisms.
  4. It is important to discuss in the separate part (like “3.5”) the other “chemical defense mechanism” (i.e. antibacterial substances) in the egg white and yolk before coming to the part “4. Conclusions”.
  5. It is important to include more strong recommendations in the part “4. Conclusions”.
  6. Moderate English changes required.

Reconsider after major revision.  

Author Response

Dear reviewers:

Thank you for your comments concerning our manuscript entitled “Antibacterial properties of TMA and effect of temperature and storage duration on TMA content in egg yolk” (ID: foods-1572810). Those comments are all valuable and very helpful for revising and improving our paper, as well as the important guiding significance to our researcher. We have studied comments carefully and have made a correction which we hope meet with approval. Revised portions are marked up using the “Track Changes” on the manuscript. The main correction in this manuscript and the responses to reviewer’s comments are below:

Response to Reviewer 3 Comments:

-------------------------------------------------------------------

Dear Authors,

the present manuscript ("Antibacterial properties of TMA and effect of temperature and storage duration on TMA content in egg yolk ") is an important and actual work. It is in a frame of the major journal scopes. I do not doubt the technical quality of the work and feel that there is a sufficient impact on a broader readership.

It will be very useful, if the authors can provide some corrections in the text and more extended discussions of the presented experiments.

Major comments:

Comment #1:

  1. It is important to correct a title in the following ways: 1) to use “Trimethylamine” instead of “TMA”; 2) there are many data on lysozyme activity and content (in addition to the TMA content in egg yolk) that must be addressed in the title too (i.e. “lysozyme activity”).

Response 1: Done. We have revised the title to “Antibacterial properties of TMA against Escherichia coli and effect of temperature and storage duration on TMA content, lysozyme activity and content in eggs”.

Comment #2:

  1. It is important to divide Figure 1. (A) and (B) on the separate Figures 1 and 2 in order to increase the size ( at least in 3 times) of the photos “(B) Schematic of the 156 plate-based assay used to assess the effect of TMA on E. coli colonies”. It will be a better indication of these significant difference.

Response 2: Done. Please see line 163-169.

Comment #3:

  1. The part “3.2. Effects of temperature and storage duration on TMA content and TBC in egg yolks” (lines 198-203). It is important to discuss the obtained TMA differences (“Changes in the TMA content in egg yolks are shown in Table 1”). It is especially important to discuss the “increased from 2.369 μg/g to 2.989 μg/g at 25°C and from 2.369 μg/g to 2.995 μg/g at 4°C” (lines 199-201), because the authors insisted that “no microorganisms were found in the yolks throughout the experiment” (lines 202-203). Usually, such changes in the TMA content in eggs must be due to the action of microorganisms.

Response 3: Trimethylamine (TMA) is a kind of biogenic amine. Biogenic amines are small molecules that are formed through decarboxylation of amino acids and through biosynthesis and enzymatic processes (Figueiredo, T.C.; Viegas, R.P.; Lara, L.J.; Baião, N.C.; Souza, M.R.; Heneine, L.G.; Cançado, S.V. Bioactive amines and internal quality of commercial eggs. Poult. Sci. 2013, 92, 1376-1384.). The increase of trimethylamine in egg yolk might be form through the processes above. Please see the discussion in line 233-241.

Comment #4:

  1. It is important to discuss in the separate part (like “3.5”) the other “chemical defense mechanism” (i.e. antibacterial substances) in the egg white and yolk before coming to the part “4. Conclusions”.

Response 4: Done. Please see line 353-374.

Comment #5:

  1. It is important to include more strong recommendations in the part “4. Conclusions”.

Response 5: Done. Please see line 375-384.

Comment #6:

  1. Moderate English changes required.

Response 6: Thanks. We have carefully edited the entire manuscript and the manuscript has been polished by a professional company. The proof of English polishing was attached.

Sincerely,

Dr. Jiangxia Zheng and Dr. Xuefeng Shi

Round 2

Reviewer 1 Report

The manuscript has been improved significantly. All the queries raised in the previous round have been addressed properly. I am satisfied with the revised version of the manuscript. I think it may be accepted in the present form. 

Reviewer 2 Report

Thanks to the authors of the work for clarifying the doubts I had about certain important sections of the work. I have no more comments. I consider that the work is ready to be accepted in its current form. 

Best regards, 

Reviewer 3 Report

Dear Authors,

The corrections made in the present manuscript ("Enhancing ocular bioavailability of ciprofloxacin using colloidal lipid-based carrier for the management of post-surgical infection") are sufficient.  This is an important and actual work. It is in a frame of the major journal scopes.

I propose to accept this manuscript in the present form.
